# Alcohol- and Low-Iron Induced Changes in Antioxidant and Energy Metabolism Associated with Protein Lys Acetylation

**DOI:** 10.3390/ijms25158344

**Published:** 2024-07-30

**Authors:** Jesse A. Thornton, Zeynep C. Koc, Vincent E. Sollars, Monica A. Valentovic, James Denvir, John Wilkinson, Emine C. Koc

**Affiliations:** Department of Biomedical Sciences, Joan C. Edwards School of Medicine, Marshall University, Huntington, WV 25755, USAsollars@marshall.edu (V.E.S.);

**Keywords:** alcohol metabolism, iron metabolism, transferrin receptor, TfR1, ferritin, FerL, ferroportin 1, FPN1, lysine acetylation, sirtuins, antioxidant and energy metabolism, oxidative phosphorylation, liver diseases

## Abstract

Understanding the role of iron in ethanol-derived hepatic stress could help elucidate the efficacy of dietary or clinical interventions designed to minimize liver damage from chronic alcohol consumption. We hypothesized that normal levels of iron are involved in ethanol-derived liver damage and reduced dietary iron intake would lower the damage caused by ethanol. We used a pair-fed mouse model utilizing basal Lieber-DeCarli liquid diets for 22 weeks to test this hypothesis. In our mouse model, chronic ethanol exposure led to mild hepatic stress possibly characteristic of early-stage alcoholic liver disease, seen as increases in liver-to-body weight ratios. Dietary iron restriction caused a slight decrease in non-heme iron and ferritin (FeRL) expression while it increased transferrin receptor 1 (TfR1) expression without changing ferroportin 1 (FPN1) expression. It also elevated protein lysine acetylation to a more significant level than in ethanol-fed mice under normal dietary iron conditions. Interestingly, iron restriction led to an additional reduction in nicotinamide adenine dinucleotide (NAD^+^) and NADH levels. Consistent with this observation, the major mitochondrial NAD^+^-dependent deacetylase, NAD-dependent deacetylase sirtuin-3 (SIRT3), expression was significantly reduced causing increased protein lysine acetylation in ethanol-fed mice at normal and low-iron conditions. In addition, the detection of superoxide dismutase 1 and 2 levels (SOD1 and SOD2) and oxidative phosphorylation (OXPHOS) complex activities allowed us to evaluate the changes in antioxidant and energy metabolism regulated by ethanol consumption at normal and low-iron conditions. We observed that the ethanol-fed mice had mild liver damage associated with reduced energy and antioxidant metabolism. On the other hand, iron restriction may exacerbate certain activities of ethanol further, such as increased protein lysine acetylation and reduced antioxidant metabolism. This metabolic change may prove a barrier to the effectiveness of dietary reduction of iron intake as a preventative measure in chronic alcohol consumption.

## 1. Introduction

Throughout civilized human history, alcohol has been the most socially accepted addictive drug worldwide. It is estimated that alcohol consumption is the third leading cause of preventable death in the United States alone, while globally accounting for up to 4% of total mortalities [1,2]. The overall worldwide incidence of alcoholic liver disease (ALD) was recently reported as 0.208/1000 person-years, with a prevalence of 4.8%, carrying a 23.9% mortality burden among the ALD population [3]. The economic burden of ALD in the United States reported in 2022 was USD 31 billion, which has been estimated to more than double (to USD 66 billion) by 2040 [4]. The first stage of ALD develops as hepatic steatosis, or “fatty liver”, and this can progress to cirrhosis or hepatocellular carcinoma with continued alcohol intake [5,6]. Chronic alcohol use is also associated with many diseases including Alzheimer’s disease, bone disease, cancer, chronic respiratory disease, diabetes, and heart disease [2,7].

The primary site of ethanol (CH_3_CH_2_OH) metabolism is in the liver, causing it to be the organ most directly affected by chronic alcohol use and pathogenesis [8]. In the liver, ethanol is oxidized in a series of reactions, first producing the toxic intermediate acetaldehyde by alcohol dehydrogenase (ADH) in the cytosol, requiring the reduction of nicotinamide adenine dinucleotide (NAD^+^) to NADH (Figure 1). Although ADH is the main enzyme, a small amount of ethanol is also converted to acetaldehyde by microsomal cytochrome P450 2E1 (CYP2E1) and peroxisomal catalase [6,9]. Then, acetaldehyde is shuttled into the mitochondria and oxidized by aldehyde dehydrogenase (ALDH2), also requiring NAD^+^, to form acetate [8]. Acetate is converted to acetyl-CoA by acetyl-CoA synthase (ACS) before it enters the tricarboxylic acid cycle to be completely oxidized in energy metabolism in mitochondria (TCA) (Figure 1). Requirement for additional NAD^+^ for ethanol metabolism causes accumulation of acetyl-CoA and alters the cellular redox state and energy metabolism due to the changes in post-translational modifications of proteins by acetylation e-amino groups of Lys residues (Ac-K) [10,11,12]. The depletion of free NAD^+^ as a consequence of ethanol metabolism is also responsible for the reduced activity of NAD^+^-dependent class III histone deacetylases and sirtuins (SIRT1-7) [10,11,13,14,15]. SIRT1-3 are the main NAD^+^-dependent deacetylases translocated into the nucleus, cytosol, and mitochondria, respectively, regulating protein deacetylation. In addition to the regulation of their activities by NAD^+^, expressions of SIRT1-3 are modulated by chronic alcohol feeding in animals [10,11,13,14,15,16].

Part of the pathogenesis associated with chronic alcohol consumption relates to how the liver regulates systemic iron availability and the potential buildup of hepatic iron stores. In biological systems, the oxidation state of iron may be either ferrous (Fe^2+^) or ferric (Fe^3+^), representing the loss of two or three electrons, respectively. The labile iron pool of cells comprises ferrous iron, which can be incorporated (in the cytosol and mitochondria) into heme, iron-dependent proteins, as iron-sulfur (Fe-S) clusters. Ferrous iron also participates in harmful redox reactions and reactive oxygen species (ROS) generation via Fenton chemistry (Figure 1). In this regard, Fe-S clusters are essential electron carriers that catalyze the production of nicotinamide adenine dinucleotide phosphate (NADPH) metabolites in plants and algae [17] or the ferric uptake regulator system in bacteria [18].

The risk of liver disease ranging from cirrhosis to hepatocellular carcinoma is greatly increased in iron overload disorders such as hereditary hemochromatosis, an autosomal recessive disease in which excess iron is absorbed and deposited in the liver hepatocytes [19]. Chronic alcohol use desensitizes the regulation of iron uptake leading to increased iron absorption and hepatic iron overload, one of the hallmarks of alcoholic liver disease [20,21,22]. In mammals, ferroportin 1 (FPN1) is the only known iron exporter and controls systemic iron homeostasis regulated by the hepcidin-ferroportin axis [23,24,25]. In the blood, iron binds to transferrin to form diferric-Tf and is translocated into cells via transferrin receptors, TfR1 and TfR2, leading to mediated endocytosis [23,26,27,28]. Another means of iron transfer is by pinocytosis, endocytosis of liquid droplets, of ferritin in the liver [28]. The redox reactive free iron inside the cell is referred to as the labile iron pool, and it is utilized for biological processes, such as heme synthesis and electron transfer processes in oxidative phosphorylation (OXPHOS), to generate ATP. The unused iron is primarily stored in a non-reactive state within the ferritin protein [29,30] (Figure 1). In its reactive state, iron is a known mediator of oxidative stress, generating reactive oxygen species (ROS) as a product of Fenton chemistry in which free iron reacts with hydrogen peroxide to form the hydroxyl radical (^•^OH) [29]. ROS can also be generated from the metabolism of ethanol by cytochrome p450 (CYP) 2E1 to acetaldehyde, an alternative cytosolic pathway that engages under high levels of cellular ethanol (Figure 1) [7,31]. 

As described above, SIRT3 is the main mitochondrial NAD^+^-dependent deacetylase responsible for regulating the activities of OXPHOS complexes as well as the two mitochondrial enzymes essential for alcohol and antioxidant metabolism, ALDH2 and SOD2, respectively [32,33,34,35,36]. ALDH2 is an NAD^+^-dependent enzyme converting acetaldehyde to acetate in mitochondria, and its acetylation at the e-amino group of the lysine residue at position 367, K367, reduces its activity while its deacetylation by SIRT3 activates the enzyme [32]. Similarly, SOD2, the mitochondrial Mn-dependent superoxide dismutase, is activated by SIRT3-dependent deacetylation at positions K68, K122, and K130 [33,34]. Alcohol consumption increases the acetylation of ALDH2 leading to a decrease in the detoxification of acetaldehyde, subsequently leading to increased aldehyde-associated ROS generation. SOD2 scavenges ROS by catalyzing the dismutation of superoxide (O_2_^•−^) into oxygen and hydrogen peroxide, which then is converted to ^•^OH in the presence of Fe^2+^ (Fenton reaction). Therefore, alcohol-induced ALDH2 and SOD2 acetylation cause increased superoxide generation and further oxidative stress in mitochondria [10,11,37,38] (Figure 1).

The present study explored the impact of ethanol treatment in the context of normal versus reduced dietary iron to gain insight into the role that normal iron levels play in ethanol-derived liver damage. The overall hypothesis is that normal iron levels were involved in ethanol-derived oxidant stress, and thus lowering the amount of dietary iron would reduce oxidative stress in the liver resulting from chronic ethanol consumption. We initially evaluated the impact of dietary iron reduction on the iron phenotype and the impact of ethanol on hepatic stress endpoints under normal and low iron conditions. The effect of low iron intake was also investigated on ethanol-induced hepatic acetylation and changes in expression of the NAD^+^-dependent deacetylases involved in antioxidant defense mechanisms. Our findings reveal mild pathological changes in liver tissue while causing significant changes in total NAD^+^/NADH levels and antioxidant enzyme expression and activities in mice fed low iron-ethanol diets.

## 2. Results and Discussion

### 2.1. Overexpression of Transferrin Receptor 1 (TfR1) Up-Regulates Iron Intake under Low Dietary Iron Conditions

Mice were on a pair-fed regimen relative to ethanol using basal Lieber-DeCarli liquid diets with four experimental groups starting at 12 weeks of age. To test the hypothesis that the reduced dietary iron intake alleviates the impact of chronic alcohol consumption, mice received normal and low iron diets along with ethanol intake for 22 weeks (see Section 3). We first tested the clinical anemia possibly caused by dietary conditions by measuring the hematocrit levels at the time of necropsy. Results indicated that the only significant change between treatment groups was a slight decrease in hematocrits in normal-iron (NI) EtOH-fed animals relative to the control diet (Figure 2A). After confirming the normal hematocrit levels, the total non-heme iron levels were determined to measure the hepatic systemic iron availability in normal and low iron (LI) pair-fed animals in basal and chronic ethanol conditions using a previously described method [39]. As expected, non-heme iron was increased by about 35% in EtOH NI pair-fed animals compared to NI alone animals (Figure 2B). This is consistent with chronic alcohol use desensitizing the regulation of iron uptake leading to increased iron absorption and hepatic iron overload, one of the hallmarks of alcoholic liver disease [20,21,22]. Non-heme iron was reduced by about 27% and 18% in mice receiving control LI and EtOH LI diets, respectively, relative to their control NI counterparts. One of the most significant changes we observed was the 46% reduction of non-heme iron accumulation in EtOH LI animals compared to the EtOH NI pair-fed control animals. This observation clearly showed that low dietary iron intake significantly reduced the hepatic non-heme iron (*p* ≤ 0.01) accumulation in the livers of EtOH LI animals based on a two-way analysis of variance (Figure 2B). These results indicate that the experimental model was successful in replicating both increased iron intake associated with chronic alcohol use and the reduction of non-heme iron associated with low iron diets in alcohol-fed animals.

Interestingly, the ten-fold decrease in dietary iron intake caused only a slight reduction in hepatic non-heme iron levels in low-iron fed animals (Figure 2B). To determine the mechanism responsible for this slight reduction, expression of iron response element (IRE) binding proteins involved in iron storage and transfer into cells, ferritin (FerL), transferrin receptor (TfR1), and ferroportin (FPN1) were evaluated by western blot analyses of liver lysates. Representative Western blots probed with FerL, TfR1, and FPN1 antibodies and quantitation of their expression determined from multiple immunoblots are shown in Figure 2C–E. In agreement with earlier studies of expression in both NI and LI groups of alcohol-fed rats, dietary ethanol led to increased hepatic ferritin expression (Figure 2C) [24]. These observations have been shown to correlate with ethanol-mediated regulation of FPN1 and hepcidin expression, also rendering hepcidin insensitive to its normal response to increases in hepatic iron stores [23,24,25]. As expected, in response to low dietary iron levels, both LI treatment groups exhibited a marked reduction in FerL protein levels relative to their corresponding NI groups. In the LI control group, FerL was reduced by 74% relative to the NI control group (*p* ≤ 0.01) while the reduction was about 50% in the EtOH LI group relative to the EtOH NI group (*p* ≤ 0.05) (Figure 2C and Appendix A). Our observations suggest that iron directly (*p* ≤ 0.001) and ethanol indirectly (*p* ≤ 0.05) impacted the expression of the FerL subunit; however, ethanol and iron did not interact in these animal groups.

Another IRE-binding protein responsible for iron accumulation in the liver associated with alcohol intake is TfR1. TfR1 is increased in the hepatic tissue of patients with alcoholic liver disease [22,40] and 12-week chronic alcohol-fed mice [41]. In agreement with these studies, 22-week-long chronic ethanol treatment also increased TfR1 expression by 1.5-fold in EtOH NI mice relative to control NI mice (an insignificant change) and by 1.8-fold in EtOH LI mice versus control LI mice (*p* ≤ 0.05) in our studies (Figure 2D and Appendix A). On the other hand, FPN1 expression increased by about 30% (*p* ≤ 0.05) to prevent iron accumulation in NI EtOH-fed mice by exporting it out of the liver cells (Figure 2E and Appendix A). As expected, low dietary iron led to a 2.4-fold increase in TfR1 expression in LI mice relative to NI controls, and 3-fold in EtOH LI mice relative to EtOH NI mice. Based on a two-way analysis of variance, dietary iron indirectly and ethanol directly significantly affected TfR1, and these two factors interacted (*p* ≤ 0.001). Increased expression of TfR1 in response to ethanol under chronic conditions of iron scarcity is unique to this study and possibly responsible for minimal changes observed in cellular non-heme iron levels under conditions of low-iron intake.

Our observations are consistent with the low-iron diets creating an iron-poor systemic environment for the liver cells, leading to reductions in hepatic iron and, thus, hepatic ferritin. After 22 weeks of EtOH treatment, the liver iron stores were depleted sufficiently to generate a low cellular iron condition consistent with the observed increase in transferrin receptor expression (Figure 2C,D and Appendix A) and the reduction in non-heme iron (Figure 2B) seen in the control LI and NI mice. Ethanol increased transferrin receptor expression in humans and mice receiving normal dietary levels of iron [22,41]. As proposed by Suzuki et al., this mechanism would facilitate hepatic iron loading if systemic iron uptake and availability were increased due to the impact of ethanol on hepcidin secretion, as has been reported [22,42,43,44]. In our experiment, systemic iron was altered through changes in dietary availability; nevertheless, we did find that TfR1 increases were accompanied by higher mean levels of FerL and non-heme iron seen in EtOH-fed mice relative to their non-ethanol-treated basal controls at either dietary iron levels.

Further, we found that ethanol’s impact on TfR1 expression (Figure 2D, control vs. EtOH LI) occurs during conditions of relative iron scarcity (Figure 2B; note the decrease in hepatic non-heme iron in LI vs. NI controls). Under these conditions, the basal TfR1 expression of the receptor is expected to be increased via the IRP-IRE (iron regulatory protein-iron responsive element) translational mechanism [45,46], as seen when comparing control NI to LI mice. This result is intriguingly consistent with the finding that the EtOH LI-fed mice had a strong rise in FerL (Figure 2C) relative to the LI controls. This observation is possibly in response to greater iron uptake due to increasing TfR1 activity (Figure 2D) despite equivalent low dietary iron conditions and equivalent stored (non-heme) iron in EtOH LI and LI controls (Figure 2B). This lack of correlation between FerL expression and non-heme iron may reflect ferritin’s role in sequestering iron that is released from the lysosome, which may make it a more specific marker for this pool of labile cellular iron [45,47]. The slight increase in FerL and non-heme iron levels in EtOH LI groups relative to LI control groups may thus indicate a relatively small increase in labile iron between these two groups. Altogether, these observations suggest that the efficacy of dietary interventions aimed at decreasing iron availability for individuals at risk because of chronic alcohol consumption may have limitations imposed by ethanol-induced increases in transferrin receptor pathway activity.

### 2.2. Ethanol and Low-Iron Intake Cause Hepatic Stress by Reducing the Oxidative Capacity of Liver Cells

Oxidative stress occurs when alcohol metabolism and ROS generation exceed the antioxidant capacity of cells provided by the availability of NAD^+^ and glutathione (GSH) (Figure 1) [48,49,50,51,52,53]. Ethanol can also have an inhibiting effect on the transmethylation pathway, impacting the regeneration of methionine in the methionine cycle [54,55,56]. When this pathway is inhibited, it reduces the amount of the methyl donor *S*-adenosyl methionine (SAMe) available for reactions requiring methyl groups, such as DNA methylation, and the GSH synthesis through the transsulfuration pathway [57,58]. We speculate that ROS derived from ethanol consumption may lead to GSH depletion and subsequent reductions in SAMe methyl donor levels and downstream methyl donor-dependent reactions.

To evaluate the pathological impact of the ethanol and low-iron diets, histological sections of the liver were prepared and examined; all EtOH-fed samples were found to have mild steatosis, but no progression beyond this stage. There were no significant changes in body weight between diet groups; however, dietary ethanol significantly impacted the liver/body weight ratios (*p* ≤ 0.001) (Figure 3A)). The liver-to-body weight ratio was increased by more than 30% in both NI and LI EtOH groups relative to their basal counterparts, control NI and LI groups (*p* ≤ 0.05 for both groups according to Tukey’s test as adjusted for multiple comparisons), indicating that hepatic stress was present (Figure 3A).

Alcohol metabolism in hepatocytes requires NAD^+^, derived from niacin, to oxidize ethanol to acetate and then to completely oxidize acetyl-CoA to CO_2_ and water in the TCA cycle (Figure 1). The large amount of NAD^+^ used in the oxidative alcohol metabolism generates the reduced form: NADH. To determine if dietary treatments led to reductive stress, we evaluated mean total hepatic NAD^+^ and NADH levels (Figure 3B and Appendix A). Based on a two-way analysis of variance, the total NAD^+^ and NADH levels were reduced by 33% and 57% in EtOH-fed animal groups in both NI (*p* ≤ 0.05) and LI (*p* ≤ 0.0001) conditions, respectively, when compared to the control NI group (Figure 3B). This observation is in agreement with niacin malabsorption in the small intestines of alcoholic pellagra patients [59]. Interestingly, the change in reductive capacity was not significantly different in the EtOH LI group relative to the LI control, suggesting that low iron is the more important factor in reductive capacity. Consistent with this, dietary iron significantly affected the total NAD^+^ and NADH levels (*p* ≤ 0.001) irrespective of ethanol treatment. This is possibly due to the reduced tryptophan uptake in the presence of alcohol or a nonheme iron-dependent dioxygenase, 3-hydroxyanthrilic acid 3,4 dioxygenase, activity involved in the kynurenine pathway of NAD^+^ synthesis, starting from tryptophan [60,61].

Oxidative metabolism of alcohol by Cyp2E1 generates NADP^+^ and ROS in microsomes, leading to GSH depletion [48,62,63]. The availability of reduced and oxidized GSH levels was determined in all diet groups (Figure 3C and Appendix A); however, the slight reduction in EtOH NI and LI mice groups was insignificant. Another metabolite that has protective action against oxidative stress caused by ethanol consumption is S-Adenosylmethionine (SAM). SAM inhibits alcohol and lipid oxidation by chelating Fe^2+^ and preventing its autoxidation [64]. Ethanol consumption has also been associated with a decreased SAMe/SAH ratio and an overall decrease in methylation, including DNA methylation [58]. To evaluate the impact of ethanol on the methionine cycle as well as the ability to regenerate the ROS scavenger glutathione, the total hepatic SAMe and SAH levels were determined using HPLC (Figure 3D). Although the data indicate that the lower dietary iron leads to decreases in both mean SAMe levels and the SAMe:SAH ratio in EtOH-fed mice vs. controls groups, the variance was high and the results were inconclusive. This trend was not observed in NI mice fed ethanol-containing diets (Figure 3D).

### 2.3. Changes in Iron Homeostasis Regulate Protein Acetylation and Sirtuin Expression in Ethanol-Fed Mice

Alcohol metabolism generates excessive amounts of acetyl-CoA in mitochondria (Figure 1) causing hyperacetylation of e-amino groups of Lys (Ac-K) residues in a wide variety of proteins in mice liver [11,12,65]. To determine the effect of low-iron on protein Ac-K, we performed immunoblotting analyses of protein lysates obtained from control and EtOH groups fed NI and LI in their liquid diets. The protein Ac-K was detected by a pan ε-amino acetyl-Lys antibody and normalized to Ponceau S staining of nitrocellulose membranes, as shown in Figure 4A and Appendix A. Based on a two-way analysis of variance, EtOH treatment led to a substantial, though not statistically significant, 72% increase (*p* ≤ 0.17) in protein Ac-K under NI conditions, whereas the LI EtOH conditions led to a 2.9-fold increase in protein Ac-K (*p* ≤ 0.001) (Figure 4A). These results suggest, for the first time, that the reduced dietary iron intake significantly increases ethanol-derived global hepatic Lys hyperacetylation in mice. 

The lysine-hyperacetylation implies a change in the reducing capacity of hepatocytes in LI EtOH-fed mice group. One prevailing hypothesis is that the depletion of NAD^+^ through oxidative ethanol metabolism (Figure 1 and Figure 3A) yields decreased NAD^+^-dependent deacetylase activity or protein expression in NI and LI mice. In fact, the major NAD^+^ dependent deacetylases, SIRT1, SIRT2, and SIRT3, regulate reversible Ac-K of nuclear, cytosolic, and mitochondrial proteins, respectively, in alcohol-fed animals [12,14,66,67]. To correlate their expression at low-iron conditions, we performed western blot analyses using SIRT1, SIRT2, and SIRT3 antibodies in NI and LI-fed mice liver lysates. A graphical representation of the protein expression of these sirtuins from multiple gels is shown in Figure 4B–D and Appendix A. Relative protein expression was normalized to Ponceau-S staining and data were expressed as a percentage of the NI control group (Figure 4).

SIRT1 is localized to the nucleus and regulates NAD^+^-dependent deacetylation of nonhistone and histone proteins in health and disease [68]. Ethanol feeding has been shown to down-regulate SIRT1 expression and deacetylation of peroxisome proliferator-activated receptor-g coactivator-1a (PGC-1a) in mice liver [67]. However, we did not observe significant changes in SIRT1 expression between the diet groups (Figure 4B). The SIRT1 protein expression was slightly reduced in EtOH LI mice; nonetheless, it was not statistically significant. Here, it is also possible that the significantly higher acetylation observed in EtOH LI mice (Figure 4A) could be due to the changes in sirtuin activity and NAD^+^ levels rather than protein levels.

The major cytosolic NAD^+^-dependent deacetylase, SIRT2, expression is negatively correlated with the severity of alcoholic liver injury, and the liver-specific SIRT2 KO sensitizes mice to alcoholic liver injury [15]. Consistent with these observations, SIRT2 expression levels were increased in EtOH NI and LI groups as compared to their controls (Figure 4C). The SIRT2 protein expression was increased slightly in the EtOH NI group relative to the NI control group (*p* ≤ 0.05) while its expression was much higher in the LI animal group (*p* ≤ 0.05) (Figure 4C). SIRT2 maintains iron homeostasis by decreasing cellular iron export and nuclear factor erythroid 2–related factor 2 (Nrf2) destabilization [69]. Iron deficiency causes reduced cell viability in SIRT2^–/–^ mice due to reduced NRF2 and ferroportin 1 (FPN1) expression, confirming the SIRT2 requirement for iron homeostasis [69]. It is possible that the increased SIRT2 expression seen in ethanol-fed mice caused a reduction in iron export, which resulted in only a 30% decrease in non-heme iron levels observed at low iron conditions (Figure 2B). Taken together, our results suggest that increased SIRT2 expression correlated with iron accumulation in EtOH fed animals compared to their controls.

Alcohol metabolism generates acetyl-CoA to be used in the TCA cycle for energy generation (Figure 1); however, excessive acetyl-CoA in the mitochondria may lead to the modification of proteins and inhibition of several TCA cycle enzymes by Ac-K [13,36,38,70]. The mitochondrial NAD^+^-dependent deacetylase, SIRT3, is responsible for the reversible acetylation of proteins and it is the most widely studied enzyme in alcohol-induced Ac-K in mitochondria [10,11,13,14]. To determine the effect of low-iron and EtOH on SIRT3 expression, we performed western blot analyses using the NI and LI animal groups. Consistent with the observed increases in Ac-K, SIRT3 expression was reduced significantly in both EtOH groups relative to NI and LI controls (*p* ≤ 0.0001 and 0.001, respectively). On the other hand, SIRT3 levels were not significantly lower in LI control animal group relative to the LI EtOH and NI controls (Figure 4D). This novel observation, along with the increase in TfR1 expression in EtOH LI animals (Figure 2D), is consistent with the regulation of iron uptake by SIRT3 as proposed by Haigis Laboratory [71]. They have shown that the enhanced iron uptake and TfR1 expression induce cell proliferation in SIRT3 KO mice and cancer cells and tumors with reduced SIRT3 expression [71]. Therefore, the correlation between increased iron uptake and reduced SIRT3 expression in human and animal studies suggests a role for SIRT3 in carcinogenesis related to chronic alcohol consumption.

### 2.4. Low-Iron Intake Reduces Activities and Expression of Antioxidant Enzymes in Alcohol-Fed Mice

Alcohol has been shown to compromise the expression and activation of antioxidant enzymes such as Cu-Zn superoxide dismutase (SOD1) and SOD2 in the liver [6,53]. These two enzymes convert O_2_^•−^ radicals to hydrogen peroxide for further detoxification by catalase found in cytosol and mitochondria (Figure 1). The SOD1 loss causes serious liver injury in alcohol-fed rats, implying that it is responsible for scavenging ROS generated by CYP2E1 in microsomes and mitochondria [72,73]. However, the effect of alcohol on SOD1 levels at low-iron conditions has never been tested. As shown in Figure 5A and Appendix A, SOD1 expression was significantly reduced in the EtOH NI group relative to the control, and this was exasperated in the LI animal group with EtOH relative to the control LI animals (*p* ≤ 0.001) detected by western blot analyses. The reduced SOD1 expression could be due to changes in NRF2, which is an important transcriptional regulator of antioxidative cellular protection, specifically regulated by iron levels in the liver [74,75,76].

The ROS scavenging in mitochondria is fulfilled by SOD2 and alcohol modulates its activity [10,77]. There are three Ac-K sites in SOD at positions K68, K122, and K130, and the reversible acetylation of these sites is shown to be regulated by SIRT3 in an NAD^+^-dependent manner [33,34,78]. Alcohol-induced Ac-K68 and Ac-K122 lead to a 40% decrease in activity without changing SOD2 levels, causing further oxidative stress in mice [37]. We detected Ac-K at position 68 and SOD2 expression and quantified their relative expressions by western blot analyses. Although we observed a slight increase in SOD2 expression, the increased Ac-K68 was significant in both NI and LI EtOH animal groups (*p* ≤ 0.0001) (Figure 5B and Appendix A). At low-iron conditions, Ac-K68 is slightly higher; however, the increase was not as noticeable as the overall Ac-K observed in Figure 4A. It is possible that the other Ac-K residues were more affected at low-iron conditions in SOD2, such as Ac-K122 and Ac-K130, or the basal Ac-K was already higher in SOD2 at control groups (Figure 5B). Here, the increase in Ac-K residues is not only correlated to the changes in SIRT3 expression or the other deacetylases but also the NAD^+^ availability in EtOH mice groups, specifically at low-iron conditions (Figure 3B and Figure 4C).

### 2.5. Alcohol and Low-Iron Conditions Cause Changes in Mitochondrial Energy Metabolism

Alcohol metabolism starts in the cytosol and continues within mitochondria to be fully metabolized to CO_2_ and water in the TCA cycle and OXPHOS, as summarized in Figure 1. Earlier studies have revealed that chronic alcohol consumption alters OXPHOS subunit expression and activities as well as oxidative stress in animal models [79,80]. These processes require iron-containing cofactors such as cytochromes and Fe-S clusters for oxidation-reduction reactions and are considered major sites for ROS generations (Figure 1). It has also been proposed that the iron accumulation and ROS generation are the major cause of alcohol-induced liver injury [20,77,81,82]. To determine the effect of reduced iron intake on mitochondrial energy metabolism of alcohol-fed mice, we detected the expression of several OXPHOS complex subunits and activities. OXPHOS subunit, ATP5A1 (complex V), UBQCRC2 (complex III), and SDHB (complex II) protein expressions were detected by the OXPHOS antibody cocktail and normalized using Ponceau stained membranes (Figure 6A and Appendix A). The expression of complex V, III, and II subunits was not changed substantially in normal and EtOH NI and LI groups (Figure 6A). This observation is in agreement with previous studies, where alcohol consumption did not impact overall protein content while mtDNA deteriorated in aging rats [83]. Conversely, OXPHOS subunit expression and activities have been reported to be significantly reduced along with ATP production in alcohol-fed rats [79,84,85].

As discussed above, alcohol metabolism is known to modify mitochondrial proteins by acetylation and their modification is further increased in alcohol-fed SIRT3 KO mice [10,13]. To verify the effect of alcohol and low-iron conditions on OXPHOS complexes, we determined the succinate dehydrogenase and cytochrome *c* oxidase activities (complex II and IV, respectively). Complex II activity was reduced in EtOH NI and slightly increased in LI EtOH relative to the control LI; however, the changes were not statistically significant (Figure 6B). The slight increase in LI EtOH complex II was not expected, particularly at the reduced NAD^+^ and SIRT3 levels shown in Figure 3B and Figure 4D. SIRT3 regulates complex II activity by reversible acetylation of SDHA subunit at several Ac-K residues and its activity is reduced in SIRT3 KO mice [36,86].

Complex IV reflects the overall electron transfer activity since it is the final step in electron transport chain complexes (Figure 1). Similar to the other OXPHOS complex activities, complex IV activity is modulated by SIRT3-dependent reversible acetylation of the core complex IV subunit, MT-COI, and the reduced SIRT3 expression impairs complex IV activity at physiological conditions [87]. As reported previously in alcohol-fed rats, we also observed a 30% decrease in Complex IV activity in EtOH NI groups relative to control NI animals (*p* ≤ 0.001). On the other hand, the EtOH LI group had relatively higher activity than the control LI group (*p* ≤ 0.01). The 20% increase in EtOH LI was the most surprising finding amongst these groups (Figure 6C). This observation agrees with the increased complex II activity in EtOH LI mice group (Figure 6B). However, we expected to observe reduced complex II and IV activities at low NAD^+^ levels as well as the reduced SIRT3 expression in EtOH LI mice due to a possible increase in acetylation (Figure 3A and Figure 4C).

## 3. Materials and Methods

### 3.1. Reagents

Sodium acetate and trichloroacetic acid (TCA) were from Fisher; 3-(2-Pyridyl)-5,6-di(2-furyl)-1,2,4-triazine-5′,5″-disulfonic acid disodium salt (ferrozine), thioglycolic acid, and hydrochloric acid 37% (HCl) were from Sigma-Aldrich (St. Louis, MI, USA); Fe iron standard 1000 μg/mL 2% HCl was from High Purity Standards (Charleston, SC, USA). Glycine, tris, glycerol, and ammonium persulfate were from USB (Cleveland, OH, USA). Triton X was purchased from Bio-Rad (Richmond, CA, USA). Acetic acid and sodium deoxycholate were purchased from Thermo-Fisher (Rockford, IL, USA).

### 3.2. Animal Studies

Male C57BL/6 mice were purchased at 10 weeks of age from Jackson Laboratories (Bar Harbor, ME, USA) and housed in an AAALAC-accredited facility and maintained on a 12 h on and 12 h off light cycle in a temperature-controlled room. Our study sought to investigate the impact of iron scarcity on the effects of ethanol in adult mice over a 22-week period. We chose to begin experimentation at the age of 10 weeks to ensure mice were beyond the juvenile stage and clearly adults. Mice were acclimatized for one week prior to starting any experimental studies. Mice were then adjusted to basal Lieber-DeCarli liquid diets [88] and fed ad-lib for one week prior to the onset of treatment. Mice were divided across two axes of treatment, normal iron (NI, 55 mg/kg of ferric citrate) vs. low iron (LI, 5 mg/kg ferric citrate) and ethanol (EtOH) vs. normal controls, generating four experimental groups of mice (control NI, EtOH NI, control LI, and EtOH LI). The ethanol content of the diets for mice in the two ethanol groups started at 2.5% of dietary kcals and was increased by 2.5% every half week until they reached 30% after 6 weeks of feeding. The ethanol diets comprised 35% calories from fat, 17% from carbohydrates, 18% from protein, and 30% from ethanol with nutritionally adequate amounts of vitamins, minerals, and methyl donors such as folate and choline. Basal control diets were identical with the exception that the calories derived from ethanol in EtOH-based diets were derived instead from maltose dextran. Diets were prepared daily from powdered diet mixes and maltose dextran (Dyets, Bethlehem, PA, USA). The ethanol used for diet preparation was Everclear (151 proof, Luxco, St. Louis MO, USA), purchased locally. Mice were pair-fed diets for 22 weeks, with individual basal NI or LI control mice receiving an equal number of calories of basal NI or LI diet as were consumed by their individual EtOH NI or LI counterparts eating EtOH NI or LI diets on the previous day. Mice were provided enrichment through housing objects and special bedding that enabled shredding and nest-building activity. Mice were monitored for morbidity through daily observation, weekly weight measurement, and daily measurement of food intake.

After 22 weeks of feeding, the mice were weighed and then euthanized by a combination of controlled exposure to carbon dioxide gas to render mice unconscious and fully unresponsive to palpebral and pedal reflex testing, followed by exsanguination by blood draw from the vena cava. The necropsy to remove organ material, which immediately followed, also included piercing the diaphragm. Hematocrits were measured and excised livers were washed in PBS, weighed, frozen in liquid nitrogen, and stored in a −80 °C freezer. Liver sections were preserved for histology analysis in buffered formalin.

### 3.3. Ethics Statement

All animal protocols were approved by the Marshall University Institutional Animal Care and Use Committee. The use of animals in these studies was under the regulations and guidelines determined by the Association for Assessment and Accreditation of Laboratory Animal Care, which accredits our institution. Our PHS Assurance and Accreditation Numbers are A3578-01 and 16-00348, respectively. Mice were given enrichment aids to counter the stress of the single housing imposed by our pair-fed design, monitored for signs of morbidity, and all efforts were made to minimize any suffering or stress.

### 3.4. Non-Heme Iron Assay

Nonheme iron was analyzed based on the colorimetric ferrozine method described by Rebouche et al. [39]. Approximately 20 mg of tissue was placed in boil-proof microcentrifuge tubes and incubated overnight at 95 °C in a heat block. Dried tissue was weighed and diluted with protein precipitation solution (0.5N HCl, 5% TCA) added at a ratio of 50 μL per mg of tissue. The tissue suspensions were mixed with vortexing and incubated at 95 °C for 1 h. The tubes were cooled in a water bath at room temperature for 2 min, vortexed, and centrifuged at 14,000 rpm for 10 min and supernatants reserved for chromogenic evaluation.

For sample preparations, 50 µL of supernatant diluted with an equal volume of distilled water was reacted with 100 µL of chromogen solution (0.508 mM ferrozine, 1.5 M sodium acetate, and 1.5% thioglycolic acid). For standards, iron concentrations of 0, 100, 200, 300, 400, 600, 800, and 1000 μg/dL were prepared in 0.25N HCl and 2.5% TCA solution and reacted at a 1:1 ratio with 100 µL of chromogen solution. After 30 min of incubation at room temperature, the absorbance was measured at 562 nm using a SpectraMax 340 by Molecular Devices (Sunnyvale, CA, USA). Samples were assessed in triplicates and standards in duplicates.

### 3.5. Western Blots

Protein from the whole cell lysate of liver tissue frozen at −80 °C was quantified using the bicinchoninic acid (BCA) protein assay kit (Thermo Scientific, Rockford, IL, USA), and subsequently separated by SDS-PAGE on 12% polyacrylamide gels. The proteins were transferred using a wet transfer system, the Mini Trans-Blot Cell (Bio-Rad, Richmond, CA, USA), onto Whatman 0.1 µm pore size nitrocellulose membranes (GE Healthcare, Little Chalfont, UK). Even transfer and equal loading of proteins were evaluated by Ponceau-S (Sigma) staining and densitometry.

Following transfer and staining, the membranes were blocked for one hour in a tris-buffered saline-Tween 20 (TBS-T) (Bio-Rad) solution containing 5% (*w*/*v*) powdered milk and 2% (*w*/*v*) bovine serum albumin (BSA) (USB, Cleveland, OH, USA). The membranes were then probed with primary antibody overnight at 4 °C. Blots were washed in TBS-T and incubated for 1 h with appropriate secondary horse radish peroxidase (HRP)-conjugated antibody at room temperature. Signals were visualized using Super Signal West Pico chemiluminescent substrate (Thermo-Scientific, Waltham, MA, USA), with 5 min incubation time and exposed to clear blue X-ray film (Thermo-Scientific) in a Kodak Biomax cassette (Rochester, NY, USA). The Ponceau S staining of nitrocellulose membranes was used to normalize total protein loading to signal intensities detected by immunoblotting analyses. Bands were quantified with densitometry and normalized to the protein stains using Un-Scan-It graph digitizing software (Silk Scientific Inc. Orem, UT, USA).

### 3.6. Antibodies

The following primary antibodies were used: FerL (New England Peptide, Gardner, MA, USA); TfR1 (Invitrogen, Camarillo, CA, USA); N-acetyl lysine (Immunechem, Burnaby, BC, Canada); SIRT1 and SIRT2 (Thermo-Scientific); OXPHOS antibody cocktail, SDHA, SOD2-K68Ac, and Ferroportin1 (FPN1) (Abcam, Eugene, OR, USA); NDUSF2 and TFAM (Santa Cruz, Dallas, TX, USA); SIRT3, SOD1, and SOD2 (Cell Signaling Technologies, Danvers, MA, USA); β-tubulin (ProteinTech, Rosemont, IL, USA); and GAPDH (Fitzgerald, Acton, MA, USA). The secondary anti-rabbit and mouse HRP-conjugate antibodies were obtained from Pierce (Rockford, IL, USA).

### 3.7. NAD^+^/NADH Assay

NAD^+^ and NADH levels were measured using the NAD^+^/NADH quantification kit purchased from Biovision (Milpitas, CA, USA). Homogenate was immediately filtered using Amicon Ultra 10 kDa centrifugal filters, purchased from Millipore (Billerica, MA, USA), to remove any NAD degrading enzymes. The assay was carried out on the resulting filtrate following the kit instructions.

### 3.8. Determination of Hepatic Glutathione and SAMe Levels

GSH was determined using a spectrophotometric assay. For the assay, 200 mg of the liver tissue was homogenized in 5% sulfosalicylic acid (SSA) for a final volume of 1 mL. GSH was assessed using glutathione reductase with 5,5′-dithiobis(2-nitrobenzoic acid) and NADPH. GSSG was measured by derivatizing the samples with 2-vinylpyridine first allowing the determination of a percentage of oxidized glutathione within the cell.

A 200 mg aliquot of the liver was homogenized on ice in 0.4 mM HClO4 and adjusted to a final 1 mL volume. Mitochondrial and 15,000× *g* supernatant suspensions were added to an equal volume of 0.4 mM HClO4 to precipitate protein. Nuclear samples were concentrated by lyophilizing the 1 mL sample (total liver weight 600–900 mg) and reconstituting the sample in 125 μL 0.4 mM HClO4. The samples were then centrifuged at 10,000× *g* at 4 °C and filtered through 0.45 μM Millex^®^-HV filters (Millipore; Billerica, MD, USA). A 20 μL sample of the filtrate was analyzed for whole liver, mitochondrial, and 15,000× *g* supernatant fractions, while 40 μL of the sample was required for the detection of nuclear SAMe levels. SAMe and SAH levels were detected using a Beckman Coulter HPLC system (Fullerton, CA, USA) with a 126 Solvent Module and a 166 Variable Wavelength Detector. The column was a YMC ODS-AQ 3 μm 120 Å 4.6 × 150 mm column. The mobile phase was a gradient at a flow of 1 mL/min (Waters Corporation; Milford, MA, USA). The mobile phase gradient program was 8 min of 90:10 A:B followed by 12 min of 60:40 A:B. Mobile phase A consisted of 8 mM 1-heptane sulfonic acid sodium salt and 50 mM sodium phosphate monobasic (pH 3). Mobile phase B was 100% HPLC-grade methanol. The wavelength for detection was 254 nm.

### 3.9. OXPHOS Complex Activity Assays

Liver tissues obtained from control and EtOH NI and LI animal groups were lysed in a buffer containing 300 mM Mannitol, 20 mM sodium phosphate, pH 7.2, 10 mM KCl, 5 mM MgCl_2_, and 2 mg/mL dodecyl-β-D-maltoside. Protein concentration was determined by BCA assay. Lysates were pre-incubated in a buffer containing 300 mM Mannitol, 20 mM sodium phosphate, pH 7.2, 10 mM KCl, 5 mM MgCl_2_, 50 mM sodium succinate, 40 mM sodium azide, prior to the addition of 50 μM 2,6-dichloroindophenolate to fully-activate the succinate dehydrogenase (complex II). Complex II enzymatic activity was recorded by monitoring the reduction of 2,6-dichloroindophenolate at 600 nm as described previously [36,89].

For complex IV activity assays, protein lysates were prepared by sonication in an assay buffer (10 mM Tris-HCl, pH 7.0, 120 mM KCl, 250 mM sucrose, 1 mM n-Dodecyl-β-D maltoside) containing phosphatase and protease inhibitor cocktails. The liver lysates for the complex IV enzymatic activity assay were diluted in a 50 mM phosphate buffer (pH 7.4) containing 1 mM EDTA and 100 µM reduced cytochrome *c*. The complex IV activities were measured spectrophotometrically by monitoring the oxidation of cytochrome *c* at 550 nm and determined as previously described [89,90].

### 3.10. Statistics

Based on our 2 × 2 study design and the need to make four comparisons between certain groups, a two-way ANOVA was used to determine the impact of the dietary factors iron and ethanol and the interactions of these factors. Significant differences between individual groups (NI vs. LI controls, NI control vs. ethanol NI, LI control vs. ethanol LI, ethanol NI vs. ethanol LI) were determined using Tukey’s test adjusted for multiple comparisons. GraphPad Prism 10 (La Jolla, CA, USA) software was used to make graphs and perform statistical calculations.

## 4. Conclusions

Alcohol consumption has been shown to elevate iron stores and cause alcohol-induced liver disease. To test the effect of low-iron intake, we investigated the changes in alcohol and antioxidant metabolism in mice. Low iron diets predictably led to decreases in ferritin expression and increases in TfR1 expression. Ethanol treatment caused a further increase in TfR1 expression at normal and low iron conditions. Additionally, ferritin expression was increased in the EtOH LI group relative to the LI controls, indicating that normal iron homeostasis may be disrupted by ethanol in a low-iron setting. The results of this study suggest that the increased hepatic iron stores in chronic alcohol consumers may involve alterations in iron homeostasis which could interfere with dietary management of iron loading through a strategy of reduced iron consumption.

In addition to the effect of low-iron intake on cellular iron homeostasis, we report that ethanol consumption has unexpected effects on NAD^+^ levels and protein Ac-K in the LI mice group compared to NI mice groups. Measurement of oxidative stress markers, reduced GSH and methyl donor levels SAMe/SAH ratio, were slightly changed between EtOH and control groups, with no significant association to dietary iron intake. After 22 weeks of ethanol treatment, low-iron intake was associated with decreased total cellular NAD^+^/NADH levels and SIRT3 expression and increased ethanol-derived cellular protein lysine hyperacetylation. On the other hand, the major cytosolic NAD^+^- dependent deacetylase, SIRT2, was increased while the expression of SIRT1 was not changed significantly. Therefore, it is possible that SIRT3 was responsible for the observed impact of the low-iron diet on protein hyperacetylation in the EtOH LI mice group. Reduced SIRT3 expression may have also assisted with cellular iron uptake by increasing TfR1 expression in the EtOH LI animal group as shown in SIRT3 KO mice studies [71]. Moreover, the increased SIRT2 expression in the LI group may decrease the cellular iron export as supported by earlier studies performed in SIRT2 KO mice [69]. Concurrently the reduced SIRT3 and the increased SIRT2 expression could be the potential mechanism to increase iron availability for cellular processes at low-iron conditions. The changes observed in antioxidant enzymes, SOD1 and SOD2, responsible for ROS scavenging were also noteworthy in the alcohol-fed mice group. Acetylation of SOD2 also leads to a decrease in activity, resulting in increased superoxide and further oxidative stress [10,37]. For this reason, the decrease in SOD1 expression and the increase in SOD2 Ac-K68 at EtOH LI condition need further attention to consolidate the roles of SIRT2 and SIRT3 iron-dependent ROS generation.

The difference detected in mitochondrial protein expressions, mainly OXPHOS subunits, was not observed in our animal groups. Thus, the reduced complex II and IV activities EtOH NI group agreed with earlier studies. However, slightly elevated complex II and IV activities in EtOH LI groups were unexpected. We speculated that the discrepancy between OXPHOS subunit expression and complex II and IV activities is due to the changes in acetylation or other post-translational modifications of proteins at low-iron conditions. Two important pathways converging at lysine modifications, acetylation and ubiquitination (Ub-K), are critical regulatory mechanisms controlling protein stability and proteosome-mediated degradation [91]. The majority of the Ac-K residues detected in proteins overlap with Ub-K sites as determined by proteomics studies [92,93]. The inhibition of proteosome-mediated degradation of the complex II subunit, SDHA, promotes oxygen consumption and increases ATP levels [94]. These findings imply that alcohol-induced Ac-K modifications can stabilize OXPHOS complexes by increasing their activities and possibly reducing their ROS-driven modifications at low-iron conditions.

Our data and earlier alcohol studies suggest that alcohol consumption regulates energy and ROS metabolism by changes to the Lys acetylation status of mitochondrial proteins. Consequently, it is essential to consider exploring the role of alcohol consumption on mitochondrial biogenesis regulated by protein Lys acetylation. The discovery of small-molecule modulators of protein Lys acetylation and enzymatic activities of Sirtuins are imperative to minimize the impact of alcohol on mitochondrial metabolism.

## Figures and Tables

**Figure 1 ijms-25-08344-f001:**
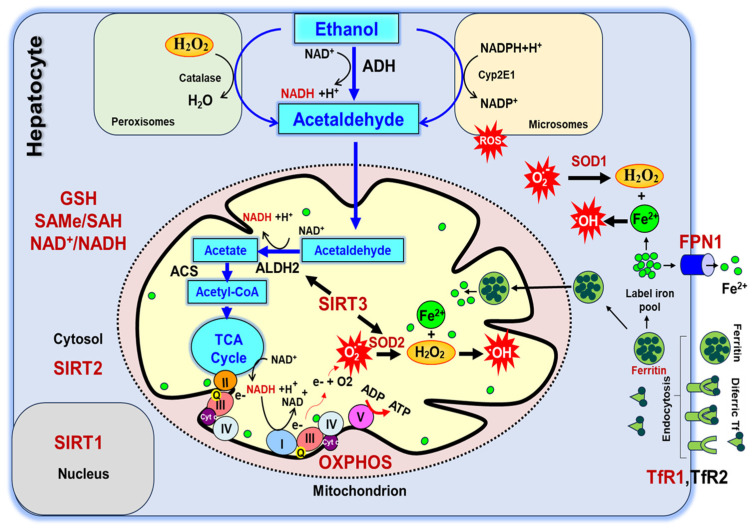
**Ethanol and Iron metabolisms converge with antioxidant mechanisms in hepatocytes.** Ethanol is metabolized to acetaldehyde in the cytosol by alcohol dehydrogenase (ADH), cytochrome p450 2E1 (Cyp2E1), and catalase. Acetaldehyde is converted to acetate and acetyl-CoA by mitochondrial aldehyde dehydrogenase (ALDH2) and acetyl-CoA synthetase (ACS), respectively. Acetyl-CoA is oxidized to generate CO_2_ and reduce NAD^+^ in the tricarboxylic acid (TCA) cycle. Electrons (e^-^) extracted from NADH are transferred to electron transport chain complexes (I, II, III, and IV), ubiquinone (Q), and cytochrome c (Cyt c) to produce ATP (complex V) through oxidative phosphorylation (OXPHOS). Superoxide dismutase 1 and 2 (SOD1 and SOD2) metabolize O_2_^•-^ radicals to hydrogen peroxide, and ^•^OH radicals are generated in Fe^2+^ excess. Ferropotin 1 (FPN1) plays an essential role in the export of Fe^2+^ (green circles) from cells to blood. Two iron atoms bind to transferrin (Tf) in the plasma as diferric atoms, and they are taken into hepatocytes by endocytosis through transferrin receptors 1 and 2, TfR1 and TfR2, as well as ferritin complexes. The iron is either stored as ferritin complexes or used in hemoglobin and iron-containing enzyme synthesis in hepatocytes. The expression of metabolic and antioxidant proteins and metabolite levels analyzed in our study are shown in red.

**Figure 2 ijms-25-08344-f002:**
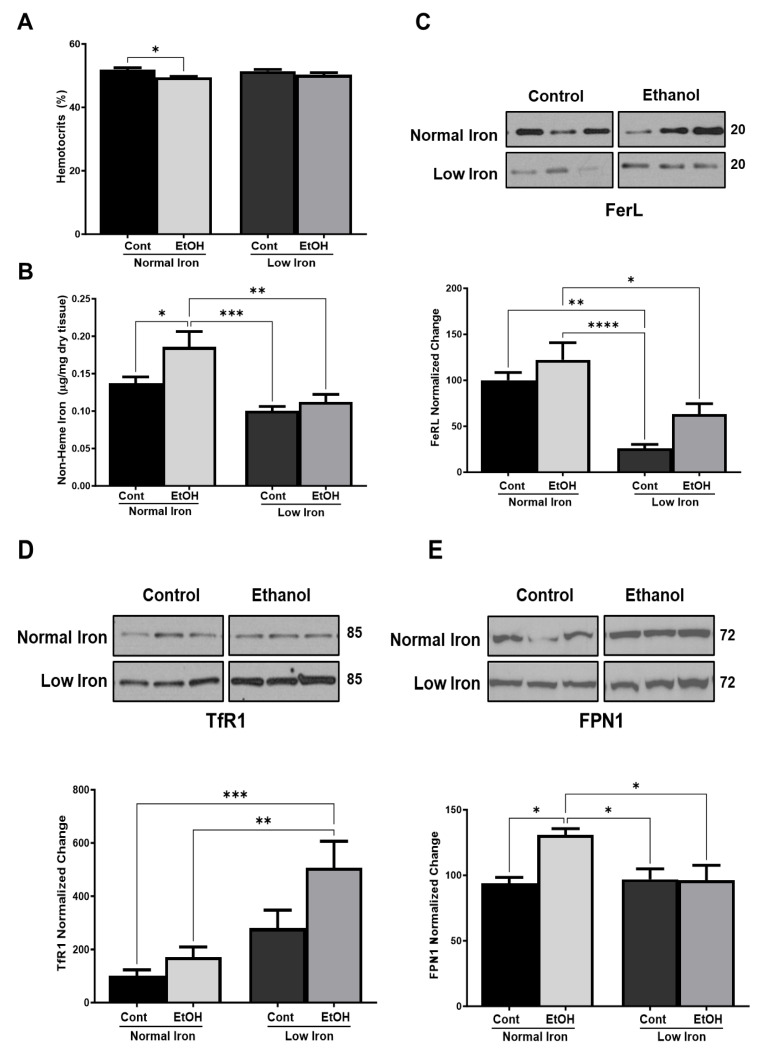
**Impact of normal and low dietary iron on iron phenotype endpoints in mice**. (**A**) Hematocrit data was taken from control and ethanol-fed mice at the time of sacrifice. Dietary groups are referred to in the text by acronyms control NI (Cont, Normal Iron), EtOH NI (EtOH, Normal Iron), control LI (Cont, Low Iron), and EtOH LI (EtOH, Low Iron). (**B**) Mean hepatic non-heme iron concentration (n = 7–8 animals assessed per group). (**C**–**E**) Expressions of FerL, TfR1, and FPN1 proteins, respectively, were detected by Western blot analyses in liver lysates. n = 4–8 animals assessed per group. Error bars represent SEM. A two-way ANOVA was performed to examine the interactions of iron and ethanol followed by Tukey’s post-hoc analysis to examine differences between groups. Stars represent: * (*p* ≤ 0.05), ** (*p* ≤ 0.01), *** (*p* ≤ 0.001), and **** (*p* ≤ 0.0001).

**Figure 3 ijms-25-08344-f003:**
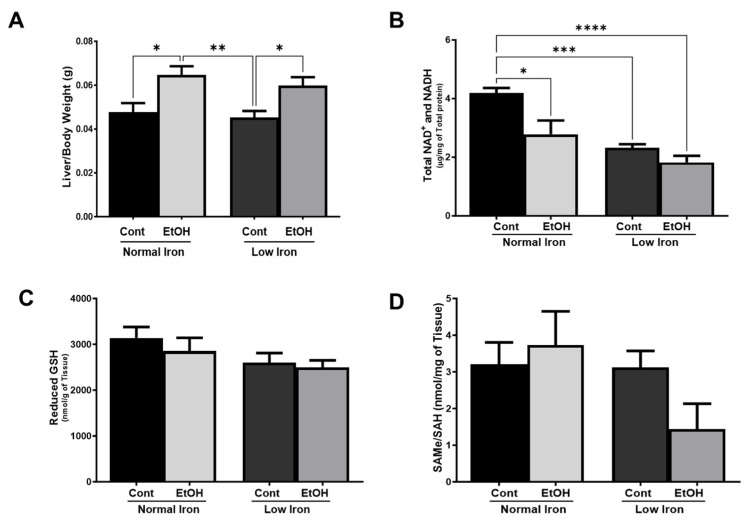
**Changes in metabolites involved in hepatic oxidative stress in mice**. (**A**) Mean liver-to-body weight ratio measured at the time of necropsy (n = 8 animals assessed per group). (**B**) Mean total hepatic NAD^+^ and NADH (n = 6–8 animals assessed per group). (**C**) Mean reduced hepatic glutathione (n = 6–8 animals assessed per group). (**D**) Mean total hepatic SAMe:SAH ratio (n = 4 animals assessed per group). Error bars represent SEM. A two-way ANOVA was performed to examine the interactions of iron and ethanol followed by Tukey’s post-hoc analysis to examine differences between groups. A two-way ANOVA was performed to examine the interactions of iron and ethanol followed by Tukey’s post-hoc analysis to examine differences between groups. Stars represent: * (*p* ≤ 0.05), ** (*p* ≤ 0.01), *** (*p* ≤ 0.001), and **** (*p* ≤ 0.0001).

**Figure 4 ijms-25-08344-f004:**
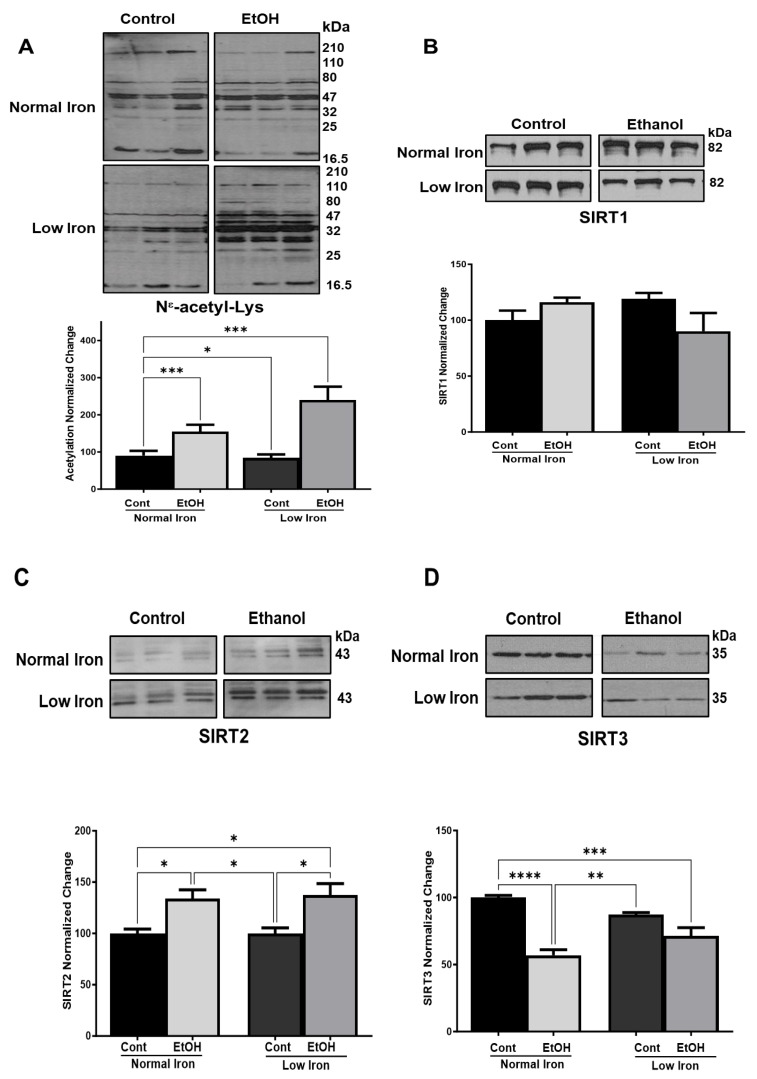
**Alcohol-induced protein lysine acetylation and Sirtuin expression at normal and low-iron conditions**. (**A**) Total protein lysine acetylation was detected by a pan N-Acetyl-Lys antibody using 20 μg of protein lysates obtained from mice fed at normal and low-iron conditions in Cont and EtOH diet groups. Densitometry of mean global hepatic Ac-K was normalized to the normal iron control group (n = 7–8 animals assessed per group) and Ponceau S-stained protein blots. (**B**–**D**) Expressions of SIRT1-3 proteins were detected by western blot analyses and the protein expression levels were normalized to the total protein loading (Appendix A). Error bars represent SEM. A two-way ANOVA was performed to examine the interactions of iron and ethanol followed by Tukey’s post-hoc analysis to examine differences between groups. Stars represent: * (*p* ≤ 0.05), ** (*p* ≤ 0.01), *** (*p* ≤ 0.001), and **** (*p* ≤ 0.0001).

**Figure 5 ijms-25-08344-f005:**
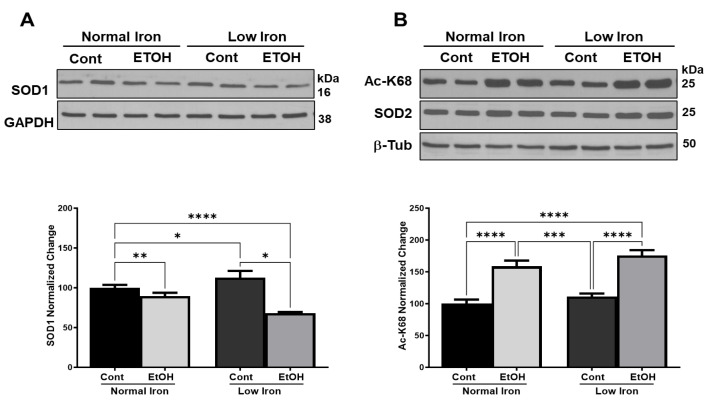
**Detection of antioxidant enzymes, SOD1 and SOD2, and SOD2 Ac-K68 acetylation in alcohol-fed animals at normal and low-iron conditions.** (**A**). Cu-Zn superoxide dismutase (SOD1) expression was detected in control (Cont) and alcohol-fed (EtOH) mice groups at normal and low-iron conditions by western blot analysis. (**B**). Mn superoxide dismutase (SOD2) expression and its acetylation at Lys68 (Ac-K68) were detected by western blot analyses in all the animal groups. Changes in Ac-K68 normalized to SOD2 detection and protein loading. Approximately 20 μg protein lysates were separated on SDS-PAGE. Relative protein expressions were normalized to GAPDH, β-tubulin (β-Tub), and total protein loading. Error bars represent SEM. Stars represent: * (*p* ≤ 0.05), ** (*p* ≤ 0.01), *** (*p* ≤ 0.001), and **** (*p* ≤ 0.0001).

**Figure 6 ijms-25-08344-f006:**
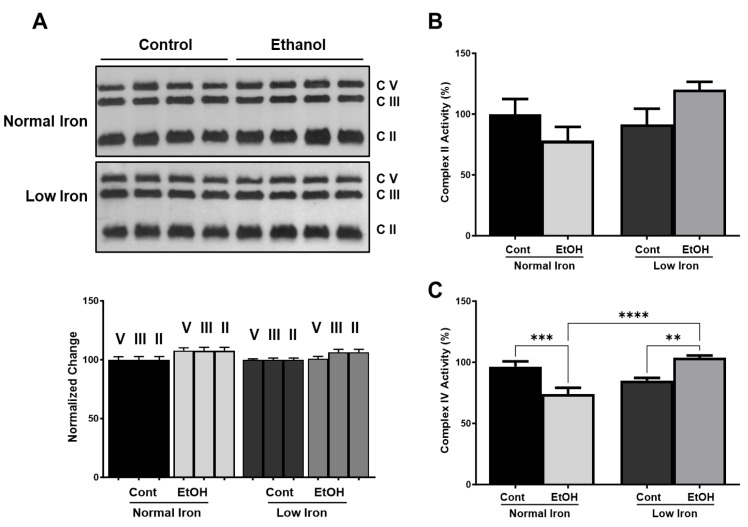
**OXPHOS complex subunit expression and activities at normal and low-iron conditions.** (**A**). Expressions of OXPHOS complex subunits, ATP5A (V), UQCRC2 (III), and SDHB (II), were detected in control (Cont) and alcohol-fed (EtOH) mice groups at normal and low-iron conditions by western blot analysis. Approximately 20 μg protein lysates were separated on SDS-PAGE. Relative protein expressions were normalized to total protein loading detected by Ponceau S staining. (**B**,**C**) Complex II and IV assays were performed as described in Section 3, and the relative activities were normalized to protein amounts and the normal iron control group values. Error bars represent SEM. Stars represent; ** (*p* ≤ 0.01), *** (*p* ≤ 0.001), and **** (*p* ≤ 0.0001).

## Data Availability

Please contact ECC for data concerning this project.

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
