# Peer review of "Alcohol- and Low-Iron Induced Changes in Antioxidant and Energy Metabolism Associated with Protein Lys Acetylation"

_ijms, 2024, doi:10.3390/ijms25158344_

Round 1

Reviewer 1 Report

Comments and Suggestions for Authors

A surprising finding by the authors. 

My concerns or points for potential improvement of the manuscript are:

The introduction is very large, I feel going to the scientific GAP the authors are addressing can be achieved in a few paragraphs and most of the text can be replaced into the discussion.

I really miss liver slides to conform the effect of the treatments at histological level preferentially with Oil-Red-O stainings to highlight lipid accumulation.

What were the negative controls in the Westm blot, and for SORT being a nuclar protein was a nuclear protein used as laoding control?

Figure 3 does not shown GSSG levels, this is important as a reduction in GSH can also be caused by leakage of GSH and not by GSSG formation. 

Can the authors give some context on the selection of the few anti-oxidant pathways investigated, why this selection and what others could be involved too?

Reviewer 2 Report

Comments and Suggestions for Authors

In this manuscript the authors describe a complex in vivo experiment with long time alcohol abuse in mice to examine the effect of low iron diet on the liver. Chronic alcoholism cause iron accumulation in the liver, so the question is correct whether iron restriction in the diet may prevent this phenomenon. The result of these experiments show, that low iron in the diet may help to reduce non-heme iron content of the liver, and increases the activities of certain electron transport complexes, but not all the effects were helpful.

The introduction gave enough information for the understanding. The experiments are planned and described correctly. The consequences of the experiments may need some further explanations; the paper needs some additional experiments.

Iron metabolism regulation is complex, the different factors may override each other. The authors mention the role of hepcidin. Though they have the liver tissue for analysis, they did not check the expression levels of hepcidin at different situations to compare the effect of alcohol abuse and iron restriction.

Also ferroportin expression could be detected to see the role of ferroportin in cellular iron accumulation.

Total NAD/NADH was determined in the experiments, but we cannot see the ratio of them, though that can be more important in the regulations.

The activities of respiratory Complex II and IV were determined in the different animal groups. What is the reason, that those two respiratory complex activities were measured?

Minor mistake: in Figure 3 C the figure legend claims to show reduced and oxidized glutathione, on the figure only reduced is labeled. The ratio of them could be also informative.

Reviewer 3 Report

Comments and Suggestions for Authors

The manuscript titled “Alcohol- and Low-Iron Induced Changes in Antioxidant and Energy Metabolism Associated with Protein Lys Acetylation” by Thornton, J.A.; et al. is a scientific work where the authors assessed the impact of dietary iron in the expression of tranferrin receptor 1 and how it increases the protein lysine acetylation levels. For it, many complementary techniques were devoted in this research. The most relevant findings could open new gates in the design of the next-generation of therapies exploiting the molecular targets against liver diseases. The manuscript is generally well-written and this is a topic of growing interest.

However, it exists some points that need to be addressed (please, see them below detailed point-by-point) to improve the scientific quality of the submitted manuscript paper before this article will be consider for its publication in the International Journal of Molecular Sciences.

1) ABSTRACT. “Interestingly, (…) in NAD+ and NADH levels” (lines 19-20). Please, the authors should define the full-name of those terms that appear for the first time in the main manuscript body text. Then, the abbreviation should be placed between brackets. “Consistent with (…) superoxide dismutase 1 and 2 levels, SOD1 and SOD2, (…)” (lines 20-24). Here, it may be desirable if the authors could placed the abbreviation between brackets instead of commas.

2) KEYWORDS. The  authors should consider to add the term “liver diseases” in the keyword list.

3) INTRODUCTION. “Throughout civilized human history (…) alcohol consumption is the third leading cause of proventable death in the United States (…) developos as hepatic steatosis, or “fatty liver”, and this can progress to cirrhosis or hepatocellular carcinoma (…) and hear disease” (lines 35-42). The authors should provide quantitative data details about the worldwide incidence burdens and the economical impact of liver diseases caused by alcohol intake. This will significantly aid the potential readers to better understand the significance of this devoted research.

4) “The primary site of ethanol (….) in which excess iron (…) regulation of iron uptake (…) liver disease (lines 43-84)” Here, even if I agree with this statement provided by the authors it would be beneficial to expand the discussion to other systems where iron plays a pivotal role in the cellular homeostasis. In this framework, iron-sulfur clusters (Fe-S) are essential electron carriers that catalyze the production of NADPH metabolites in plants and algae [1] or the ferric uptake regulator system in bacteria [2].

[1] Pérez-Domínguez, S.; et al. Nanomechanical Study of Enzyme :Coenzyme Complexes: Bipartite Sites in Plastidic Ferredoxin NADP+ Reductase for the Interaction with NADP. Antioxidants 2022, 11, 537. https://doi.org/10.3390/antiox11030537

[2] Hou, C.; et al. Revisiting Fur Regulon Leads to a Comprehensive Understanding of Iron and Fur Regulation. Int. J. Mol. Sci. 2023, 24, 9078. https://doi.org/10.3390/ijms24109078

5) MATERIALS & METHODS. 2.2. Animal Studies “Male C57BL/6 mice were purchased at 10 weeks of age from (…)” (lines 142-144). Why did the authors select mice with this age range and not younger specimens? A brief statement should be provided in this regard.

6) “2.9. OXPHOS Complex Activity Assays” (lines 254-270). Did the authors employ specific inhibitors to dissect specific components of the electron transport chain? Some information should be provided in this regard.

7) RESULTS & DISCUSSION. Figure 2 (line 307). It may be opportune if the authors could plot the Figures 2-6 in colour (I guess there is no fee-cost in this journal). This will enhance the visibility of the results found in this research.

8) “Consistent with this, dietary iron significantly affected the total NAD+ and NADH levels (…) due to the reduced tryptophan uptake in the presence of alcohol or a nonheme iron-dependent dioxygenase, 3-hydroxyanthrilic acid 3,4 dioxygenase, activity involved in the kynurenine pathway of NADP+ synthesis starting from tryptophan” (lines 413-417). Even, if I agree with this statement provided by the authors, it should not be neglected the contribution of the acetic acid cycle and how the aconitase and succinate dehydrogenase enzymes aid in the electron transfer contributing to NADH production and the subsequent homeostasis.

9) CONCLUSION. This section perfectly remarks the most relevant outcomes found by the authors in this work. The authors should add a brief statement to discuss about the future line actions to pursue this research and the open perspectives.

Comments on the Quality of English Language

The manuscript is generally well-written albeit it may be desirable if the authors could recheck it in order to polish those final details susceptible to be improved.

Round 2

Reviewer 2 Report

Comments and Suggestions for Authors

I accept the answers and the changes made on the manuscript.